# Mitochondrial Functioning and the Relations among Health, Cognition, and Aging: Where Cell Biology Meets Cognitive Science

**DOI:** 10.3390/ijms22073562

**Published:** 2021-03-30

**Authors:** David C. Geary

**Affiliations:** Department of Psychological Sciences, University of Missouri, Columbia, MO 65211-2500, USA; GearyD@Missouri.edu; Tel.: +1-573-882-6268

**Keywords:** cognitive aging, dementia, cognitive ability, mitochondria, mitochondrial dysfunction, oxidative stress

## Abstract

Cognitive scientists have determined that there is a set of mechanisms common to all sensory, perceptual, and cognitive abilities and correlated with age- and disease-related declines in cognition. These mechanisms also contribute to the development and functional coherence of the large-scale brain networks that support complex forms of cognition. At the same time, these brain and cognitive patterns are correlated with myriad health outcomes, indicating that at least some of the underlying mechanisms are common to all biological systems. Mitochondrial functions, including cellular energy production and control of oxidative stress, among others, are well situated to explain the relations among the brain, cognition, and health. Here, I provide an overview of the relations among cognitive abilities, associated brain networks, and the importance of mitochondrial energy production for their functioning. These are then linked to the relations between cognition, health, and aging. The discussion closes with implications for better integrating research in cognitive science and cell biology in the context of developing more sensitive measures of age- and disease-related declines in cognition.

## 1. Introduction

Performance in one cognitive domain, such as attentional control, is positively correlated with performance in all other cognitive domains, such as reading comprehension, and performance in all of these domains is correlated with current and predictive of later health outcomes. These relations suggest a common biological mechanism that contributes to cognition and health; moreover, this mechanism has been linked to systematic and parallel declines in cognition and health with normal aging [1,2,3]. Mitochondrial functioning, including contributions to cellular energy production, control of oxidative stress, immunity, and intracellular signaling (among others), is well situated to explain at least some of these links [4,5], as represented in Figure 1. Indeed, mitochondrial dysfunction contributes to the cognitive declines (e.g., memory loss) associated with age-related diseases, such as Alzheimer’s disease [6,7], but the links are broader than this. A focus on mitochondrial functioning provides a means to better integrate research in cell biology and cognitive science, and in doing so will expand our understanding of the fundamental biological mechanisms that underlie brain and cognitive development and functioning and result in more sensitive assessments of age- and pathology-related changes in cognition. 

The first section provides a brief overview of the relations among cognitive abilities, such as reasoning and general knowledge, and discussion of how mitochondrial functioning might contribute to these relations. The second section overviews the relation between age-related changes in cognition and health and shows that the mechanisms that contribute to the positive correlations among cognitive abilities also contribute to the relation between cognition and health. The implications for assessing age-related changes in cognition and common age-related neurological disorders are discussed in the third section.

## 2. Cognition and Mitochondrial Functions

### 2.1. Cognitive Abilities

It was discovered more than a century ago that performance on perceptual (e.g., tone detection), cognitive (e.g., memory span) and academic abilities are positively correlated, leading Spearman to conclude that “that all branches of intellectual activity have in common one fundamental function (or group of functions)” [1] (p. 285), which he termed general intelligence. The pattern of positive correlations among sensory, perceptual, and cognitive abilities is called the positive manifold and is one of the most replicated findings in the behavioral sciences [8]. Most of the associated studies, however, were conducted with Western samples, leaving the universality of the pattern in question. A recent review of 97 related studies across 31 non-Western countries and that included more than 50,000 people confirmed the positive manifold is universal and explains 46% of the covariance among cognitive measures [9]. In other words, about half of the variability (i.e., individual differences) in cognitive performance is due to a mechanism that is common to all cognitive abilities and the remaining variance is specific to that ability (e.g., reading comprehension) or is measurement error.

The relations among cognitive abilities can be placed in a hierarchy, which typically has three levels. Figure 2 shows the two higher levels where the circles represent clusters of highly correlated abilities. For instance, executive functions refer to the cognitive mechanisms that manage, direct, or organize other processes [10,11], and are composed of related but distinct abilities that would be part of the lower level of the hierarchy (not shown). The lower-level components of executive functions include working memory (the ability to hold one thing in mind while engaged in another mental activity), inhibition (suppression of task-irrelevant information), and shifting (ability to shift from one task, back to another). Performance in each of these areas is dependent on strong top-down attentional control [11,12].

The key finding is that these second-order ability domains are positively correlated, such that individuals with higher performance on measures of executive functions also have above-average performance on reasoning tasks and have more general knowledge, in keeping with a mechanism or mechanisms common to all cognitive domains. There has been an active search for these mechanisms since Spearman’s 1904 publication [1], and the associated proposals range from a mathematical artifact that emerges from statistical approaches for analyzing the relations among cognitive measures [13,14]; to basic cognitive processes, e.g., speed of processing fine-grain detail [15]; to systems of brain regions [16]; and to more fundamental biological process, e.g., control of oxidative stress [3,17].

The bulk of recent research has focused on complex cognitive systems associated with, for instance, executive functions and imaging studies of the underlying brain regions [16,18,19,20,21]. The application of network analyses to the results from these brain-imaging studies have substantively advanced our understanding of the relations between cognition and patterns of brain activity. One potential interpretation of the results of these studies is that there is a common system of brain regions that is engaged during all cognitive processing, such as those associated with attentional focus [21]. Though appealing and consistent with theories of human cognition [22], such studies do not directly address why these same mechanisms are also correlated with a wide range of health outcomes [3] or the parallel age-related declines in cognition and health [23,24].

These latter patterns are consistent with the existence of an even more fundamental mechanism (or mechanisms) that links cognition to health and aging, which is sometimes called a general biological fitness factor [25] or body integrity [3]. These arguments follow from the empirical results but leave unanswered the question of what biological processes underlie general fitness or body integrity. Hill and colleagues proposed that mitochondrial functions are the cornerstone of biological fitness [26,27], and in the next section, I elaborate on how their model can be expanded to explain the relations between cognition, health, and aging [5].

### 2.2. Nested Mechanisms

The cognitive abilities, such as attention-demanding problem solving, that show the most decline with cognitive aging [28], and with age-related cognitive pathologies (e.g., Alzheimer’s disease) are supported by intermodular brain networks. These networks are hierarchal in organization, with the functioning of higher-level systems dependent on the integrity of lower-level systems, as shown in Figure 3 [5,29,30]. In theory, the energy demands associated with the development, maintenance, and optimal functioning of these systems increase exponentially as one moves from lower (e.g., individual neurons) to higher (i.e., intermodular systems) network levels [31].

From this perspective, energy is central to the organization, maintenance, and functioning of all complex networks [31,32], which places the biological substrates and mechanisms of energy production as the linchpin of brain and cognitive functioning. The efficiency of mitochondrial energy production is a critical part of this linchpin [5]; in particular, the pathway that creates the most adenosine triphosphate (ATP), that is, oxidative phosphorylation (OXPHOS) within mitochondria. Variation in energy production might then contribute to individual differences in the functioning of cognitive systems and age-related change in energy production is a potential mechanism contributing to cognitive aging (Section 3.2). Among other things, deficits in the substrates (e.g., pyruvate) that fuel OXPHOS, dysfunction in electron transport and ATP production, and variation in the latent capacity of mitochondria to increase ATP production, among other factors, can contribute to individual differences in energy production and age-related intra-individual changes in energy production [33].

To put the importance of mitochondria in perspective, consider that the adult human brain consumes about 20% of the body’s metabolic energy at rest, largely to maintain excitatory neurons in a ready state [34]. These resources maintain about 86 billion neurons and an estimated 164 trillion synapses [35,36]; it has been estimated that each neuron consumes about 4.7 billion ATPs per second at rest [37,38], although this will vary with the number of synapses [34]. The energy requirements of individual neurons can increase 3.5-fold with the propagation of an action potential [39,40], and are heightened further with the synchronized activity of distributed brain regions (i.e., intermodular functions) associated with complex cognition [41]. Energy drops could then lead to degradation of network cohesion, that is, loss of fine-tuned small-world (intramodular) and long-range (intermodular) connections supporting core cognitive competencies.

As noted, complex cognition is supported by long-distance white matter connections between different brain regions [20,21], such as the integrated activity of the anterior cingulate, dorsolateral prefrontal cortex, and superior parietal cortex during attention demanding problem solving [16,18]. This executive control network contributes to the reasoning and executive functions competencies shown in Figure 2 [30]. Functional and resting state magnetic resonance imaging (MRI) studies have identified other intermodular systems that are common across people, some of which are evolutionarily conserved [42]. An example is provided by the dorsal attention network that orients the individual to external space and integrates visuospatial attention with sensory and motor systems, as well as more localized systems for processing sensory information (e.g., visual information) and integrating motor responses [42,43,44].

Another is the default mode network that is involved in integrating the emotional and motivational state of the individual with self-referential thoughts and memories of past experiences. The network is active during relaxed states and results in self-relevant reflections about past memories and future goals and provides “a self-centered predictive model of the world” [45] (p. 443). Although evolutionarily ancient, there are uniquely human features of the network. For instance, the precuneus (Figure 4) is involved in feelings of agency, self-awareness, personal memories, and thinking about the world in ways that involve the self [46,47], and other areas, such as the medial prefrontal cortex, are especially important during conscious reflections about the self, including explicit goal-directed self-evaluations [48]. The default mode network, in combination with the executive and attentional control network, is involved in the explicit problem solving that enables people to generate a conscious representation of themselves in the context of past and future social scenarios, among other contexts [49]. These scenario building competencies are important for navigating social dynamics [50].

The functioning of each of these intermodular brain networks is in turn dependent on small-scale networks within these brain regions and recent studies confirm the importance mitochondrial functioning, especially energy production, to the integrity of these systems. Liu et al. [52] matched the dynamic integration of large-scale networks, obtained through brain imaging, and gene-expression profiles of several participants who later died and donated their brains to the project. One finding was that the expression of genes that influence mitochondrial functioning predicted functional connectivity between prefrontal and parietal cortices, which include aspects of the executive control and default mode networks. This was confirmed in a similar study, whereby genes associated with mitochondrial energy production were highly expressed in brain regions that support long-distance intermodular networks but were less important for the functioning of small-scale intramodular networks [53]. In other words, the functioning of intermodular networks that integrate distributed brain regions, as with the default mode network, is highly dependent on cellular energy production and thus disruptions in energy production will be most evident in these systems and in the cognitive abilities that are supported by them.

## 3. Cognitive Aging and Health

Mitochondrial functions provide a natural link between cognition, health, and aging [54]. One reason is the 10^5^ mitochondria in the mammalian oocyte are randomly distributed among the cell lines that will form different physical systems [55]. Deleterious or salubrious mitochondrial DNA (mtDNA), as well as nuclear DNA (nDNA) that influence mitochondrial functioning, will be distributed throughout the body, creating the potential for similarities in the relative functioning of high-energy systems, such as the heart and brain. It is not that straightforward, however, as tissue-specific configurations of mitochondria eventually emerge, potentially related to tissue specific expression of nuclear genes that influence mitochondrial functioning [56,57]. Still, the initial pool of mitochondria and supporting mtDNA and nDNA will be highly similar across all physical systems and thus could easily contribute to a general fitness factor or body integrity [3,25,27,58]. The underlying mechanisms, including mitochondrial functioning, provide a ready link between health and brain and cognitive functioning.

The mitochondrial link extends to age-related changes in health and cognition, as Harman [4] proposed decades ago and for which there is growing support [59,60,61,62]. In broader context, aging may be an unavoidable consequence of the evolution of eukaryotic cells and the ability to upregulate energy production [63]. The benefits that contributed to the evolution of enhanced energy production likely included more rapid somatic development early in life and advantage in reproductive competition and parental investment in early adulthood, but come at a cost of gradual decline of the energy-producing capacity of these same systems [63,64]. The decline results from, among other things, the excess production of reactive oxygen species (ROS) during OXPHOS and inadequate control of ROS levels. Low levels of ROS are important for intracellular signaling but higher levels can damage DNA (especially mtDNA), cell membranes, and proteins needed for many biochemical processes [33,65,66]. The result is a slow, age-related degradation in mitochondrial functions, including reduced capacity to produce cellular energy. The slow degradation of mitochondrial functions should result in parallel declines in health and cognition with normal aging in adulthood and this appears to be the case.

In the sections that follow, I review the non-intuitive link between individual differences in cognitive abilities and individual differences in various health indices and their relation to aging. The focus in these sections is on the evidence for a single mechanism or group of related mechanisms that contribute to the links between health and cognition and that contribute to age-related declines across cognitive abilities.

### 3.1. Cognition and Health

A positive relation between general health and cognitive ability was documented almost a century ago [67] and replicated in 1992 by Lubinski and Humphreys [68] but was not systematically studied until the seminal work of Deary and his colleagues [69]. In the latter, the cognitive ability and family background of the entire school population of Scotland was assessed (across several cohorts) at age 11 years and their health was tracked over the next 68 years (the study is ongoing). The childhood measure included a mix of verbal, reasoning, spatial, practical, and arithmetic items and thus provided a broad assessment of cognitive ability.

The results of this study and similar ones show that cognitive ability assessed in childhood predicts health and risk of premature mortality throughout the lifespan [3,70,71,72,73]. In one of these analyses, the relation between age 11 cognitive ability and the odds of living to age 79 years was assessed for 70,805 participants [74]. For each one standard deviation increase in cognitive ability, there was a 20% reduction in the odds of dying before the age of 79. Forty-five percent of the individuals in the bottom 20% of childhood cognitive ability survived to age 79, as compared to 65% of individuals in the top 20%. Other analyses revealed that cognitive ability is related to diverse aspects of physical health, ranging from handgrip strength [75] to risk of coronary disease [76,77].

The relation between cognitive ability and various health measures remains with control of childhood and current socioeconomic status [3,78]. Iveson and colleagues [73], for instance, showed that childhood cognitive ability predicts premature mortality independent of the SES of the family of origin, although lower family of origin SES also predicts risk of premature mortality above and beyond cognitive ability. Moreover, the relation between cognition and health remains positive and significant with control of health-related behaviors, such as smoking and alcohol abuse [3].

The proposed mechanisms underlying the relation between cognition and health include general cardiovascular fitness, exposure to toxins (e.g., through cigarette smoking), brain metabolism, and oxidative stress [3,17], among others. In a review of the associated literature, Deary and colleagues concluded that “This idea [body integrity], which is often rather vaguely articulated … demands a search for other possible markers of system integrity—other measurable indicators of bodily and brain efficiency” [3] (p. 63). Mitochondrial functioning is a strong candidate as a key underlying mechanism linking health and cognition [76,79,80], but testing this hypothesized link is not straightforward.

If mitochondrial functioning is a critical mechanism linking health and cognition, then cognitive deficits should be common in individuals with mitochondrial disorders. These disorders are, however, quite varied in severity and rate of progression [81,82], and as a result, a definitive understanding of any associated cognitive deficits remains to be achieved. Current studies suggest these deficits are especially pronounced for complex, attention-demanding, cognitive processes (e.g., executive functions); are more common in symptomatic (e.g., having seizures) than asymptomatic individuals; and become progressively worse as the disease progresses [83,84].

Additional evidence for a direct link between mitochondrial functions and cognition comes from studies of disruptions to the typically homeostatic availability of energy substrates (lipids, glucose). These disruptions can be illustrated by the over availability of substrates resulting in obesity and physical exercise that consumes them [85]. The key idea is that obesity-related disruption of glucose homeostasis is associated with cognitive declines and accelerated aging that are at least in part related to compromised mitochondrial functions and that reductions (e.g., through exercise) in excess energy substrates improves cognition and may slow the aging process.

The many ways in which obesity can compromise mitochondrial functions are described elsewhere [85,86,87,88,89]. A definitive link between obesity, mitochondrial dysfunction, and cognition remains to be forged but there is suggestive evidence, including moderate (Effect Size, ES ≈ 0.3 to 0.45) differences in components of executive functions across obese and normal weight individuals. These patterns are likely bidirectional, with poor executive functions associated with difficulties in regulating food intake, and eventual compromises in executive functions resulting from disruptions in mitochondrial functions [88,90]. For instance, Spyridaki et al. [91] found lower reasoning abilities in obese as compared to normal weight individuals that appeared to be mediated by inflammation, one indicator of compromised mitochondrial functioning. Prospective studies confirm a relationship between chronic inflammation that can result from and further damage mitochondria and cognition [92,93,94,95]. Interventions that restore glucose homeostasis and reduce inflammation and mitochondrial dysfunction may improve cognitive functioning. The interventions range from substantive weight loss to insulin administration to pharmaceutical reductions in oxidative stress to vigorous exercise [89,96,97,98]. These interventions can be helpful, but individuals differ in their responsiveness for reasons that are not fully understood.

In any case, a recent meta-analysis included estimates of longitudinal change in cognition following weight loss and the effects that emerged in randomized controlled trials (RCT) for weight-loss interventions [99]. The longitudinal studies suggested weight loss and exercise are associated with moderate to strong gains in attentional control, executive functions, and memory abilities [ESs = 0.30 to 0.66]. However, these effects could be inflated because the repeated assessments involved in longitudinal studies often result in improved test performance without underlying cognitive gains. The RCTs, however, revealed that a combination of diet and exercise results in improved attentional control (ES = 0.44), memory (ES = 0.35), language (ES = 0.21); there were no effects for executive functions, but this was assessed in only two studies and thus the null result is not reliable. 

In keeping with a direct relation between mitochondrial functions and cognition, the authors of the meta-analysis [99] concluded that the cognitive gains might be related to reductions in insulin resistance and improvements in glucose metabolism, as well as reductions in inflammation and oxidative stress, among other things.

### 3.2. Cognition and Aging

As noted, declines in mitochondrial functioning across adulthood are the evolutionary trade-off that comes with the ability to ramp up energy production during development and during the reproductive demands of early adulthood [61,63]. It is not simply metabolic rate that eventually undermines mitochondrial functions (e.g., through oxidative stress and associated damage to mtDNA) but also mechanisms for the control of oxidative stress, among other factors [59]. The point here is that if mitochondrial functions contribute to the links between health and cognition, both health and cognition should decline in parallel with normal aging in adulthood, which seems to be the case.

Psychologists have been studying age-related changes in cognition for many decades and declines in speed of executing various cognitive processes have been repeatedly demonstrated [100,101,102], as have declines in more complex abilities, such as reasoning and executive functions [2,103]. Critically, these studies have revealed a common mechanism that contributes to declines across cognitive abilities and that seems to be directly linked to biological aging [104,105]. Based on results from the Berlin Aging Study, Lindenberger and Baltes concluded that there appears to be a common mechanism influencing age-related declines in reasoning, visual and auditory acuity, and sensory-motor skills, e.g., balance; [23], suggesting “aging changes in the physiological state of the brain” [106] (p. 352).

Tucker-Drob and colleagues’ [2] meta-analysis of longitudinal studies revealed that a common factor explained about 60% of the variation in age-related changes in cognition from middle-age to old-age. In other words, there were common processes underlying age-related declines across cognitive domains. The importance of this factor increased from middle-age (explaining 45% of the covariation among cognitive abilities) to age 85 years (explaining 70%). Moreover, this common factor was moderately correlated (*r* = 0.49) with the same mechanisms that are common to performance across cognitive domains in young adults, as was shown in Figure 2. They concluded that “individual differences in human cognitive abilities may have an inherent structure along which growth and decline naturally occur” [2] (p. 294).

There are, in addition, secular changes in the rate of cognitive aging and in various indicators of physical health that appear to be related to the Flynn effect. The latter refers to cross-generational increases in performance on cognitive tests throughout the 20th century [107,108,109]. There are likely many factors that contributed to this effect. These include increases in years of schooling that in turn make cognitive assessments more familiar to people and thus could improve performance without substantive changes in underlying abilities. However, this is unlikely to be the whole story. The improvements in medical care and nutrition, as well as reductions in social and economic stressors, that occurred throughout the 20th century were associated corresponding gains in health, e.g., as indicated by gains in height; [110,111]. The secular declines in rate of cognitive aging are associated with these gains in health [112,113]. In other words, secular changes have concurrently improved people’s cognition and health and slowed the rate of cognitive aging. Although it is not likely to be the only factor, mitochondrial functions are part of the common denominator to all of these relations [5,58,85,114,115].

Indeed, prospective studies show a relation between risk factors that are correlated with compromised mitochondrial functioning, such as chronic inflammation, and rate of cognitive decline with aging and with health [92,93,94]. A 10-year longitudinal study of 45- to 69-year-olds illustrates the relation [95]. Here, individuals with indicators of chronic inflammation at baseline showed accelerated declines in reasoning and memory abilities relative to their healthier peers. The declines were equivalent to an additional 1.7 years of normal age-related cognitive changes over the 10 years of the study.

Cognitive aging studies have also revealed that complex abilities, such as reasoning, decline more quickly than do more basic processes, such as passive memory for words. These patterns are in keeping with the expected relation between mitochondrial functions and brain and cognitive complexity described in Section 2.2. Brain imaging studies also show that more complex brain networks, including the default mode network (Figure 4), are more susceptible to age-related declines and pathology (e.g., Alzheimer’s disease) than are smaller-scale networks [116,117,118,119,120,121,122]. Age-related declines in the coherence of large-scale networks in turn are correlated with cardiovascular fitness and this is related in part to the beneficial effects of exercise and in part to genes that influence both physical fitness and more shallow age-related declines in network coherence [116]. Findings such as these are important and in keeping with the general thesis because, as described in Section 2.2, long-range brain networks are more energy dependent than are shorter range networks and thus are more vulnerable to age-related declines in mitochondrial energy production [32,41].

### 3.3. Mitochondrial Deficits and Cognition and Aging

Studies of mitochondrial contributions to neuronal growth and functioning and their contributions to age-related pathologies, such as Alzheimer’s and Parkinson’s diseases, provide mechanistic insights into how mitochondria contribute to cognition and link cognition, health, and aging [6,7,123,124,125]. As an example, decline in the integrity of the hippocampal-dependent learning and memory system is a feature of normal aging, is accelerated in Alzheimer’s disease [7], and can be linked to changes in mitochondrial functioning [126]. Sustained presynaptic firing in hippocampal neurons and synaptic plasticity associated with learning and memory are dependent on several mitochondrial functions. At synaptic boutons, mitochondria contribute to the synthesis of neurotransmitters, provide the ATP needed for the repetitive cycling of synaptic vesicles, and modulate the intracellular Ca^2+^ levels that promote release of neurotransmitters after being triggered by an action potential [124,127]. Mitochondria are also motile and can move from one synapse to the next and in this way provide dynamic adjustments such that more active synapses are supplied with sufficient ATP to maintain increases in activity levels [123].

All of these mechanisms and others have been implicated in age-related changes in cognition, age-related brain pathologies, in deterioration in health more broadly, and are associated with the earlier-described deficits that emerge with chronic excess energy substrates [125,128]. For instance, with progressive heart failure there is disrupted intracellular Ca^2+^ modulation and mitochondrial Ca^2+^ overload that results in increased permeability of the mitochondrial inner membrane and increased production of cell damaging ROS. The latter further disrupts Ca^2+^ modulation that in turn contributes to further heart damage [129]. As noted, presynaptic mitochondrial dysfunctions contribute to normal cognitive aging, as well as the expression of neurological diseases [127]; these are associated with increases in oxidative stress, decreases in ATP production, and declining Ca^2+^ buffering mechanisms [130]. These in turn are exaggerated in highly active brain regions, including the hippocampus and at synaptic boutons [131].

Devine and Josef [127] argued that synaptic degeneration might precede the onset of neurodegenerative diseases, such as Alzheimer’s disease, and is closely related to deteriorating mitochondrial functioning, see also [130]. Amyloid-β (Aβ) plaques (misfolded peptides) are a feature of Alzheimer’s disease and their accumulation undermines mitochondrial Ca^2+^ modulation and ATP production. The accumulation of intracellular tangles of tau, a protein that can become misfolded, occurs along with Aβ plaques and disrupts mitochondrial functioning directly and impedes mitochondrial transport within neurons. One result is fewer mitochondria at synaptic boutons and premature elimination of synapses [132].

Aspects of mitochondrial functioning are sensitive to estradiol (E_2_) and thus may contribute to sex differences in cardiovascular and brain health, differences that can change with the onset of menopause and reductions in E_2_ concentrations [7]. Estradiol appears to enhance glucose transport, and upregulates antioxidant defenses that in turn will reduce the rate of age-related accumulation of damage due to oxidative stress [130]. Estradiol also increases mitochondrial tolerance of the Ca^2+^ overload that is a critical aspect of the above-noted synaptic functions [133]. These protective benefits are lost with aging. One compensatory response appears to be the catabolism of myelin to produce ketone bodies as an energy supply that in turn contributes to white matter deterioration in females and increased risk of neurodegenerative diseases [134].

### 3.4. Summation

Overall, there is consistent evidence for links between cognition and health and that these same links contribute to parallel declines in cognition and health with normal aging in adulthood and with risk of age-related pathologies. These same links are sensitive to wide-scale secular changes in nutrition, prevalence of parasitic and infectious diseases, educational experiences, and social and economic stressors. No doubt, there are many contributing factors, but one factor that is known to influence and be sensitive to all of them is mitochondrial functioning [5,85]. Studies of the mechanisms through which mitochondrial functions contribute to the links between cognition, health, and aging will lead to a fuller appreciation of the biological systems that support human cognition, as well as the discovery of biomarkers of risk of cognitive decline [135].

## 4. Implications for Study of Aging and Dementia

The study of age-related declines in cognition and its relation to changes in physical health is typically based a battery of neuropsychological tests that assess a variety of memory and problem-solving competencies, e.g., [136]. Performance on these measures is sensitive to age-related declines and presence of pathology (e.g., Alzheimer’s disease) but these studies have not been fully integrated with the findings described in Section 2.1, that is, the positive manifold (that all cognitive abilities are correlated) and with Tucker-Drob and colleagues’ [2] findings of a common mechanism or set of mechanisms that underlies cognitive aging and is correlated with performance across all cognitive measures.

To better align these literatures, studies in cognitive aging could use a weighted composite measure that captures performance across all of the measures used in any neuropsychological battery. The composite is more likely to be sensitive to age-related cognitive change than will performance on any single measure or an overall score across different types of items. This is because the weighted composite provides an assessment of whatever mechanisms are common to performance across all cognitive measures.

The construction of the weighted composite first involves the use of principle components or factor analyses to assess the extent to which each of the tests loads on a primary or first factor [137]. The associated loadings provide an estimate of the extent to which the test is correlated with whatever is common to all of the tests. For instance, returning to Figure 2, reasoning tests will typically have higher loadings than language tests and thus would be more heavily weighted in constructing the composite score. One important feature is that individual differences on these composites are highly consistent across any battery of tests that might be administered, as long as there are a number of different tests [138].

Second, if changes in mitochondrial functioning are an important aspect of the mechanisms that are contributing to age-related declines across all cognitive abilities, then performance on the composite will be particularly sensitive to changes in mitochondrial functioning [5,115]. From this perspective, within-person longitudinal changes in composite scores, as with normal age-related changes in adulthood, might then provide a global estimate of the rate of degradation from the peak functioning of all of the systems represented in Figure 3. Any such declines should be highly correlated with age-related changes in the coherence of large-scale, intermodular networks, including the default mode and executive control networks, as assessed in resting-state MRI studies [52,53,139].

Third, measures of executive attention are highly correlated with these composite scores and will be a useful addition to the study of cognitive aging. As noted by Kane and Engle [140], executive attention refers to “a capacity whereby memory representations are maintained in a highly active state in the presence of interference, and these representations may reflect goal plans, action states, or task-relevant stimuli in the environment.” [140] (p. 638). Executive attention is maintained by a large-scale intermodular network anchored by areas within the dorsolateral prefrontal cortex and individual differences in the functioning of this network, along with reasoning and problem-solving abilities, explain much of common variation across cognitive abilities [140,141,142].

In other words, as shown in Figure 3, at the level of cognitive measurement the ability to maintain attentional focus under conditions that involve distractions and to engage in multi-step problem solving explains much of the commonalities across cognitive measures. Attentional control is important for performance on all perceptual and cognitive measures and when combined with problem solving explains individual differences in competence in many real-world settings [142]. Psychometrically valid measures of attentional control (https://englelab.gatech.edu/attentioncontroltasks, accessed on 28 March 2021) and complex working memory (https://englelab.gatech.edu/complexspantasks, accessed on 28 March 2021), along with demonstration videos (https://englelab.gatech.edu/taskdemos, https://englelab.gatech.edu/taskdemonstrations, accessed on 28 March 2021) are now available and could be fruitfully added to studies of cognitive aging, along with the above noted cognitive-composite measure.

The importance of executive attention and problem-solving abilities does not mean that variation in mitochondrial functioning or age-related or disease-related changes in these functions are not important. Rather, changes in mitochondrial functioning would manifest as disruptions in the coherence of large-scale brain networks, including the network that supports executive functioning. In other words, performance on measures of executive attention and the cognitive-composite might then provide highly sensitive behavioral measures of cognitive aging and cognitive deficits associated with disorders that involve mitochondrial dysfunctions, such as those associated with disruptions to glucose homeostasis (Section 3.1). These behavioral measures might also prove to be useful for the assessment of the efficacy of pharmacological treatments of mitochondrial disorders as a means to address age-related or disease-related cognitive decline [143]. 

## 5. Conclusions 

There is now consistent evidence that various mitochondrial functions, including energy production, control of oxidative stress, and intracellular signalling, among others, contribute to the well-documented relations between cognition, health, and aging [3,56,62,85,116,118]. These relations provide a natural link between research in cell biology and cognitive science. Advances in the latter are potentially useful for the development of measures that will be the most sensitive to age- or disease-related disruptions of mitochondrial functions or therapeutic enhancement of these functions and their potential influence on cognition. Declines in the coherence of the intermodular brain networks that support the complex cognitive abilities that are the most sensitive to age- or disease-related disruption make network coherence a potentially useful biomarker of cognitive pathology, e.g., Alzheimer’s disease; [117,118,119,120,121,122]. At the same time, the link between cognition and network functioning and the corresponding energy demands of these network [31,32] provide an additional means to understand how mitochondrial functioning could be expressed in cognition [5], as well as how they might also influence brain functions related to mental health [144].

Research in cell biology, especially cellular energy production, provides cognitive scientists with potential mechanistic explanations of well-documented but poorly understood human universals, including the positive manifold (indicating common mechanisms across all sensory, perceptual, and cognitive abilities) and common factors underlying age-related declines in cognition and health [2,3,5,9]. These universals are currently understood as abstractions or latent (unobserved) factors, but there must be underlying biological mechanisms. These include, as noted, the complex intermodular systems that support complex cognition [16,18,19,20], but these systems do not explain the link between cognition and health or why cognition and health show parallel declines with aging. The common denominator for the development, maintenance, and optimal functioning of all complex systems is energy and for eukaryotic cells, mitochondrial functions are the primary source of this energy, and must therefore be a significant contributor to these human universals [5].

## Figures and Tables

**Figure 1 ijms-22-03562-f001:**
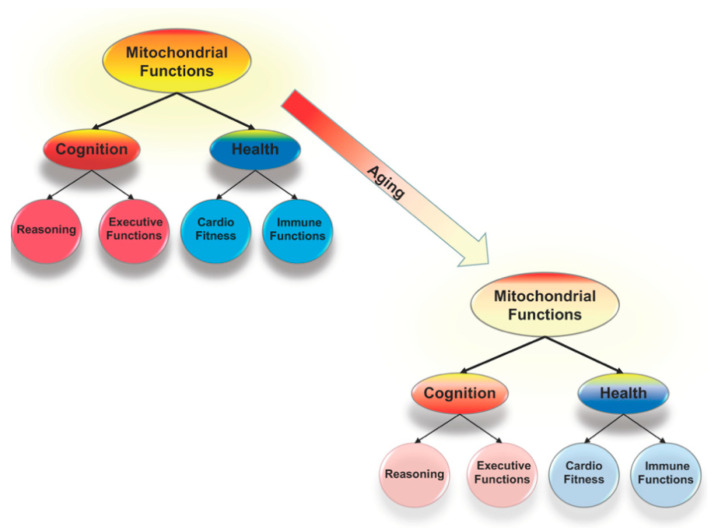
Mitochondrial functions, including cellular energy production and control of oxidative stress, are critical to the system of brain mechanisms that underlies all forms of cognition and link cognition and health. Reductions in mitochondrial functioning are an aspect of normal aging and compromise health and cognition. These reductions might be the key to understanding why there are correlated age-related declines across cognitive domains that are linked to declines in health.

**Figure 2 ijms-22-03562-f002:**
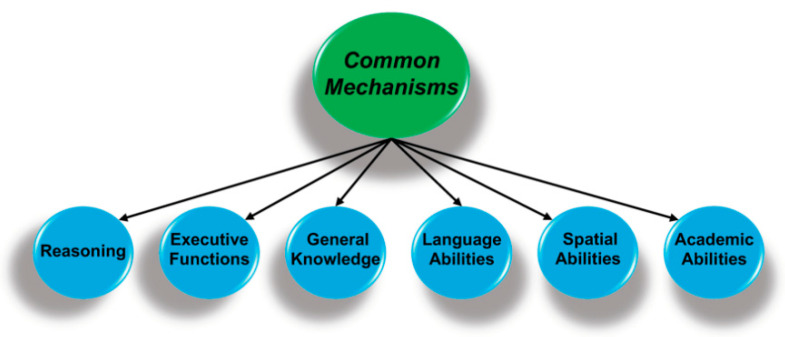
Performance on perceptual, cognitive, and academic measures are hierarchically organized, with specific abilities (not shown) at the lowest level that are organized as clusters of related abilities (above). The key finding is that performance is correlated across all domains and suggests there are mechanisms that are common to all forms of cognition.

**Figure 3 ijms-22-03562-f003:**
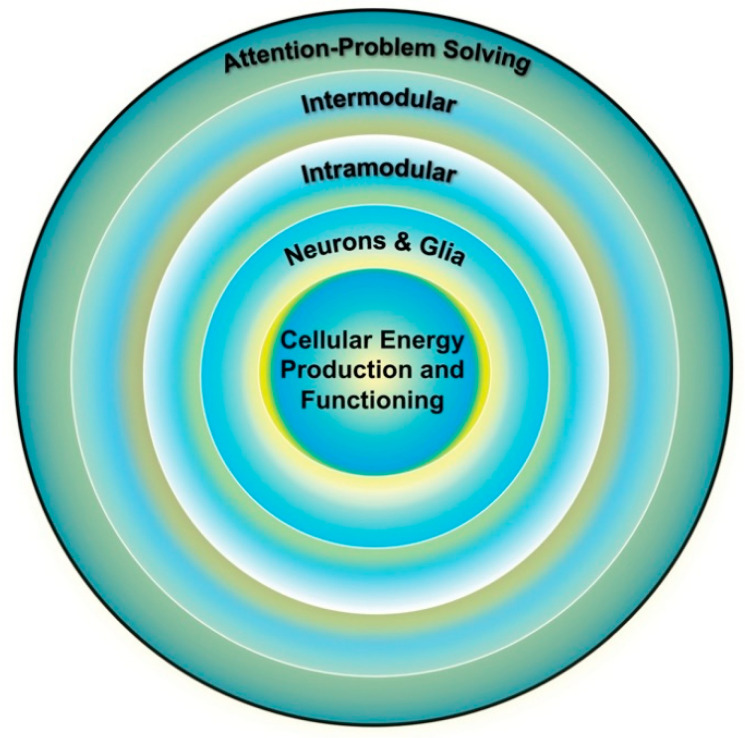
Individual differences in cognition (outer ring) are influenced by the functioning of multiple brain systems, the optimal functioning of which is dependent on the efficiency of the systems below it. Cellular energy is the lowest common currency driving the development and expression of all biological systems and thus places upper-limit constraints on the development and expression of all other systems, including cognition. Adapted from Geary [29] (p. 4).

**Figure 4 ijms-22-03562-f004:**
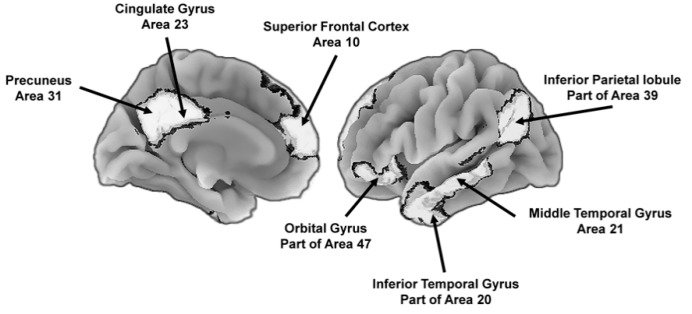
Key brain areas of the default mode network that support the construction of self-centered mental representations of the world, including potential future states. To the left is the medial (center) view of the brain and to the right is the lateral (outer side surface) view. The numbers next to the labels are Brodmann [51] map coordinates.

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
