# Peer review of "Mitochondrial Functioning and the Relations among Health, Cognition, and Aging: Where Cell Biology Meets Cognitive Science"

_ijms, 2021, doi:10.3390/ijms22073562_

Round 1

Reviewer 1 Report

The manuscript “Mitochondrial Health and Cognitive Aging: Where Cell Biology Meets Cognitive Science” is an overview of the relationship between functional mitochondria and cognition and its implications in aging. The topic is interesting, and the review is informative and well written.  It’s suitable for publication by IJMS after addressing some minor concerns from this reviewer: It’s a little confusing about the focus of the review. The title presents as “Mitochondrial Health and Cognitive Aging”, however the text focuses on that mitochondria play a role in connecting individual health, cognition and aging. Given that the health of mitochondria and the general health of an individual (the healthy status of an individual) are two different aspects, the title of the manuscript is somehow misleading, which is better to be modified.

Author Response

Thank you, this is a good point.  I’ve modified the title.

Reviewer 2 Report

The review is timely and interesting. However, the review lacks several important references. In general, molecular mechanisms need to be better discussed. The following are the specific comments:

  1. The introduction needs to be lucid for the interest of the general reader. Right now, the review is very field-specific.
  2. The review will benefit from a section discussing the molecular pathways that alter mitochondrial health affecting cognition.
  3. Please discuss how beta-amyloid and taupathy affects mitochondrial health or vice versa.
  4. Please also discuss the effect of estrogen, which is a major regulator of mitochondrial health affecting cognition.
  5. Please discuss and cite the following literatures: https://www.sciencedirect.com/science/article/pii/S2213231720304122 https://www.nature.com/articles/s41598-020-78551-4 https://www.pnas.org/content/100/5/2842.short https://pubmed.ncbi.nlm.nih.gov/23063123/ https://pubmed.ncbi.nlm.nih.gov/23891000/ https://pubmed.ncbi.nlm.nih.gov/23791179/ https://pubmed.ncbi.nlm.nih.gov/23168220/ https://pubmed.ncbi.nlm.nih.gov/23328668/ https://pubmed.ncbi.nlm.nih.gov/22772899/ 

Author Response

The review is timely and interesting. However, the review lacks several important references. In general, molecular mechanisms need to be better discussed. The following are the specific comments:

  1. The introduction needs to be lucid for the interest of the general reader. Right now, the review is very field-specific.

I’ve re-written the introductory paragraphs to make them more transparent for the general reader.

  1. The review will benefit from a section discussing the molecular pathways that alter mitochondrial health affecting cognition.

I agree and have added this to the ms; Section 3.3.  Mitochondrial Deficits and Cognition and Aging

  1. Please discuss how beta-amyloid and taupathy affects mitochondrial health or vice versa.

This has been added to Section 3.3.

  1. Please also discuss the effect of estrogen, which is a major regulator of mitochondrial health affecting cognition.

This has been added to Section 3.3.

  1. Please discuss and cite the following literatures: https://www.sciencedirect.com/science/article/pii/S2213231720304122 https://www.nature.com/articles/s41598-020-78551-4 https://www.pnas.org/content/100/5/2842.short https://pubmed.ncbi.nlm.nih.gov/23063123/ https://pubmed.ncbi.nlm.nih.gov/23891000/ https://pubmed.ncbi.nlm.nih.gov/23791179/ https://pubmed.ncbi.nlm.nih.gov/23168220/ https://pubmed.ncbi.nlm.nih.gov/23328668/ https://pubmed.ncbi.nlm.nih.gov/22772899/ 

Thank you for these links, the articles are very helpful and have been integrated into the ms (Section 3.3), along with a number of related article.